# Metabolic and Bariatric Endoscopy: A Mini-Review

**DOI:** 10.3390/life13091905

**Published:** 2023-09-13

**Authors:** Benjamin Charles Norton, Andrea Telese, Apostolis Papaefthymiou, Nasar Aslam, Janine Makaronidis, Charles Murray, Rehan Haidry

**Affiliations:** 1Digestive Diseases and Surgery Institute, Cleveland Clinic London, 33 Grosvenor Pl, London SW1X 7HY, UK; 2Department of Gastroenterology, University College London Hospital Euston Road, London NW1 2BU, UK; 3Centre for Obesity Research, Rayne Institute, Department of Medicine, University College London, London WC1E 6BT, UK; 4Bariatric Centre for Weight Management and Metabolic Surgery, University College London Hospital, London WC1E 6BT, UK; 5Department of Endocrinology and Diabetes, University College London Hospital, London WC1E 6BT, UK; 6National Institute of Health Research, UCLH Biomedical Research Centre, London W1T 7DN, UK

**Keywords:** bariatric endoscopy, gastric balloons, gastric remodelling, duodenal resurfacing

## Abstract

We are currently in a worldwide obesity pandemic, which is one of the most significant health problems of the 21st century. As the prevalence of obesity continues to rise, new and innovate treatments are becoming available. Metabolic and bariatric endoscopic procedures are exciting new areas of gastroenterology that have been developed as a direct response to the obesity crisis. These novel interventions offer a potentially reversible, less invasive, safer, and more cost-effective method of tackling obesity compared to traditional bariatric surgery. Minimally invasive endoscopic treatments are not entirely novel, but as technology has rapidly improved, many of the procedures have been proven to be extremely effective for weight loss and metabolic health, based on high-quality clinical trial data. This mini-review examines the existing evidence for the most prominent metabolic and bariatric procedures, followed by a discussion on the future trajectory of this emerging subspecialty.

## 1. Background

Obesity is fast becoming the most significant health problem of the 21st century. Worldwide, the prevalence of obesity has increased year-on-year with >650 million obese adults in 2016 [1]. An estimated 26% of men and 29% of woman are obese in the UK based on 2020 national statistics [2]. This is concerning given that obesity promotes the incidence of several serious conditions including type 2 diabetes (T2D) [3], cardiovascular disease [4], increased cancer risk [5], and many overlooked effects on morbidity including shortness of breath, back pain, reduced mobility, mental health problems, and overall reduced quality of life [6]. If rates continue to rise, this will cost society an estimated GBP 50 billion by 2050, with the NHS spending GBP 6.1 billion on overweight and obesity-related ill health in 2014–2015 [7,8].

To address this health crisis, new and innovative treatment strategies are needed, which has led to the rise of metabolic endoscopy, which is an emerging field in gastroenterology. The principal aim is to use novel endoscopic techniques as primary treatments for obesity and related complications. The technology of the devices to support these techniques has been rapidly improving, and the techniques have been shown to be highly effective for weight loss (WL) and metabolic health in high-quality clinical trials. Numerous devices exist with varying degrees of evidence to support their use, which are summarised in Table 1. Many devices are limited to small case series, others have been withdrawn due to safety concerns, and some companies have encountered financial difficulties. In this mini-review, we provide an overview on the pathophysiology of obesity and the implications of surgery that have led to the development of minimally invasive endoscopic procedures. We explore the current scientific literature on the most prominent endoscopic procedures including intragastric balloons (IGBs), gastric remodelling, duodenal mucosal resurfacing (DMR), and transoral outlet reduction endoscopy (TORe). Finally, we look at their roles as primary metabolic and bariatric treatments, and how they are beginning to shape a new subspeciality. 

### Search Strategy

We conducted a database search up to June 2023 within EMBASE and MEDLINE using relevant search terms to describe the individual procedures or devices included in this review. The search was limited to full text results in the English language. After removing duplicates, we screened 942 items for IGBs, 419 items for gastric remodelling, 64 items for DMR, and 49 items for TORe. Subsequent searches were conducted for IGBs and gastric remodelling to refine the breadth of the literature in relation to specific clinical outcomes using ‘population, intervention, outcome’ (PIO) statements allowing for inclusion of studies without a control population. 

## 2. The Pathophysiology of Obesity and Weight Regain 

At its core, weight gain occurs when energy intake chronically exceeds the net requirements of an individual. This leads to several adaptive physiological changes including altered gut hormone secretion, appetite regulation, and adipose tissue function. Gut hormones including ghrelin, glucagon-like peptide 1 (GLP-1), and peptide YY (PYY) have been identified as key regulators of energy homeostasis and eating behaviour, which are primarily released from enteroendocrine cells (EECs) within the small intestinal mucosa [34,35]. With prolonged overnutrition, excess nutrients are converted into lipids such as triglycerides and free fatty acids that are stored within adipose tissue and as ectopic fat in different organs [36]. The resulting increase in fat mass promotes oxidate stress within adipocytes and development of a proinflammatory state [37].

The key regulator of this process is insulin, which is released from beta cells of the pancreas. With excess nutrient intake, the pancreas produces more insulin in response to high circulating blood glucose levels that, overtime, can lead to insulin resistance [36]. Insulin resistance describes a condition whereby cells no longer respond appropriately to circulating insulin [38]. Overtime, the combination of hyperglycaemia, hyperinsulinaemia, and insulin resistance leads to long-term complications including type 2 diabetes and cardiovascular disease [39]. Without intervention, prolonged resistance to insulin will result in beta cell failure and the need for exogenous insulin [36]. Therefore, intervention to promote weight loss is important to reduce this proinflammatory state, to reduce insulin resistance, and ultimately, to preserve beta cell function to prevent metabolic failure and its associated complications.

One of the difficulties with conventional weight loss strategies is that they trigger powerful biological mechanisms. We know that traditional calorie-restricted diets can lead to short-term weight loss, but are associated with significant reductions in anorexigenic hormones (i.e., GLP-1) and increases in orexigenic hormones (i.e., ghrelin) that results in weight regain in the majority of people [34,40]. On the contrary, manipulation of the gastrointestinal tract during bariatric surgery results in more durable weight loss and a nearly immediate and very durable anti-diabetic effect that is weight independent [41]. These changes are thought to occur from the release of anorexigenic hormones that improve whole-body insulin sensitivity [42,43]. Unfortunately, surgery alone is not capable of addressing the magnitude of the obesity pandemic. This has led to two exciting areas of development in the treatment of obesity. First, the use of pharmacologically derived gut hormones that replicate the changes observed following bariatric surgery, and second, the development of endoluminal procedures that mimic the anatomical and physiological effects of bariatric interventions which are the focus of this review [44].

## 3. Treatment Strategies for Obesity 

Lifestyle interventions form the cornerstone of managing patients who are overweight or obese. This includes hypocaloric diets and/or increased physical activity alongside behavioural techniques to promote WL [45]. However, this strategy is not successful in all people with variable efficacy on WL. Therefore, management often requires escalation to bariatric surgery, and more recently, novel pharmacological and endoscopic options. 

### 3.1. Surgery

Bariatric surgery is the most effective treatment for patients with severe obesity, and it can lead to sustained WL, reduced mortality, and improvement in obesity-associated co-morbidities [46]. The two main procedures are Roux-en-Y gastric bypass (RYGB) and laparoscopic sleeve gastrectomy (LSG), which are both associated with sustained long-term WL (Figure 1) [47,48,49]. Bariatric surgery is associated with multiple physiological changes including increased GLP-1, reduced ghrelin, microbiome changes, and enhanced bile acid delivery [35,42,43,50,51]. These changes collectively result in WL and improved whole-body insulin sensitivity which underscores the importance of small bowel nutrient sensing. Unfortunately, surgery alone is not capable of addressing the magnitude of the obesity crisis. Multiple barriers exist including resources and scalability, operative risks, irreversibility, patient selection, and patient preference. 

### 3.2. Pharmacological

New pharmacological agents including GLP-1 agonists and combined GLP-1/gastrointestinal peptide agonists have been shown to be highly effective obesity treatments [52,53]. Across the globe, these drugs are hailed as the panacea to modern obesity management. Liraglutide (Saxena, Novo nordisk, Bagsværd, Denmark) is a GLP-1 agonist given as a once daily injection that is currently licensed for the treatment of obesity [54]. The safety and efficacy of liraglutide was confirmed within the SCALE (Satiety and Clinical Adiposity—Liraglutide Evidence) clinical trials [55,56,57,58]. Liraglutide was associated with a mean weight loss percentage of 8.0% versus 2.6% in a placebo group over 56 weeks, and the most common side effects were gastrointestinal including nausea, diarrhoea, constipation, and vomiting. [55]. Semaglutide (Wegovy, Novo nordisk, Bagsværd, Denmark) is a newer, long-acting GLP-1 agonist given as a weekly injection, which has recently been licensed for the treatment of obesity [59]. The safety and efficacy of semaglutide was confirmed within the STEP (Semaglutide Treatment Effect in People with Obesity) clinical trials, which showed a 12.4% placebo-subtracted mean weight loss percentage over 68 weeks [53,60,61,62,63]. Similar to liraglutide, the most common adverse events were gastrointestinal (e.g., nausea, diarrhoea, vomiting, and constipation), which were mostly mild and transient. 

While the above-mentioned drugs are highly effective, they involve injections, are not tolerated in all patients, have supply and cost issues, and rely on patient concordance [64]. In addition, when semaglutide was withdrawn in the STEP-1 extension study, participants regained on average two-thirds of their prior WL [65]. Therefore, additional therapies will be essential for treating obesity in the future as both an alternative option, and as part of combined therapy. 

### 3.3. Endoscopy

New endoscopic techniques for obesity offer a potentially reversible, minimally invasive, safer, and more cost-effective method of tackling obesity, which mitigates against the risk and irreversible nature of bariatric surgery [44,66,67]. These one-off interventions are highly attractive for patients who are also intolerant, or want to avoid long-term medication. The devices primarily target the stomach or small bowel to reduce weight and improve metabolic health. Many devices have been trialed over the last decade with varying degrees of efficacy, yet results with ESG have been very impressive. These devices are broadly safe with serious adverse events (SAEs) < 5%, although some procedures have shown major safety signals (e.g., hepatic abscesses in bypass liners). 

## 4. Gastric Balloons

An IGB is a space-occupying device inserted into the stomach as a primary treatment for WL. These implantable devices reduce the volume of the stomach, decrease hunger, and ultimately reduce food intake. Commercially available balloons were first developed in the 1980s with the Garren–Edwards gastric bubble (GEGB) undergoing several clinical trials; however, the balloon was limited by efficacy and SAEs including ulcers, bowel obstruction, pancreatitis, and bleeding [68,69,70,71]. Consequently, an international meeting of experts took place in 1987 leading to the Tarpon Springs Criteria that outlined key characteristics of an ideal intragastric device [68,72]. 

Today, there are several commercially available IGBs with different characteristics (Table 2). Fluid-filled balloons tend to have better efficacy, but tolerance is lower with higher rates of early removal [73,74]. The physiological mechanisms remain open to debate, but probably relate to gastric distention, ghrelin signalling, and altered gastric emptying [75,76,77]. Clinical recommendations for IGBs are limited with few guidelines. The American Gastroenterology Association recommend their use alongside moderate-to-high intensity lifestyle intervention [78]. Brazil [79] and Spain [80] have both published consensus statements for IGBs, whereas the UK National Institute for Health and Care Excellence (NICE) retain uncertainty about their safety and efficacy [81]. 

### 4.1. Effect on Weight Loss

Modern experience with IGBs spans over two decades with most evidence surrounding the fluid-filled Orbera device. Meta-analyses have confirmed consistent short-term results with a total body weight loss (TBWL) percentage of 13.2% (95% CI 12.3–14.0) at six months among 5549 patients [14], and an excess weight loss (EWL) percentage of 25.4% (95% CI 21.5–29.4) at 12 months among 1683 patients [15]. Nevertheless, long-term efficacy remains to be uncertain, with Kotzampassi et al. [82] showing only 25% of individuals maintained >20% EWL at five years and Chan et al. [83] reporting no significance difference in weight compared with controls at ten years. AEs are common with pain and nausea frequently reported, and SAEs include migration, perforation, and death in some trials [15]. Early removal due to tolerance is not insignificant, with 18.2% in one study [84]. 

Spatz is an adjustable fluid-filled balloon with 400–750 mL of saline [85]. Randomised data have shown a TBWL of 15.0% (95% CI 13.9–16.1) at 32 weeks compared to 3.3% (95% CI 2.0–4.6) with lifestyle intervention alone. SAEs occurred in 4% with no migration, pancreatitis, perforation, or death [16]. However, a randomised Brazilian cohort showed no significant difference on TBWL% with volume adjustment [86]. 

Elipse is a ‘procedureless’ balloon that does not require endoscopy for insertion and self-deflates leading to excretion [85]. It is a biodegradable device attached to a thin catheter. Once swallowed, the device is confirmed in place using X-ray or ultrasonography, and then inflated to 550 mL with liquid. The balloon stays in place for approximately four months. During this time, it automatically degrades triggering a self-releasing valve that causes the balloon to empty and be naturally excreted through the GI tract. Several prospective trials have looked at the benefit of the Elipse procedureless balloon, but most are limited to small numbers and follow-up. Ramai et al. [17] conducted a meta-analysis involving seven prospective studies and showed a pooled TBWL% of 12.2 (95% CI 10.1–14.3) at four months. Early deflation occurred in 1.8% and small bowel obstruction in 0.5%, but no other SAEs. Abdominal pain and vomiting occurred in 37.5% and 29.6% respectively, which was lower than AEs reported for other balloons. Jamal et al. [18] showed that among 90 patients undergoing Elipse insertion who reached one year follow-up, the TBWL% was 7.9%. There are no long-term data on the efficacy of the device, but within the UK, NICE have recommended it can be used for short-term WL with the provision of special arrangements [87]. 

Heliosphere is an air-filled balloon that is inserted during endoscopy and inflated to 700 mL. Early prospective data among 82 patients showed a mean TBWL of 13.0% (SD 7.0) over an average of 182 days follow-up. There were no SAEs, but early removal was seen in 1.2% and spontaneous deflation in 3%. The Obalon balloon is filled with 250 mL of a nitrogen-mix gas and two additional balloons can be swallowed over time. The randomised, sham-controlled SMART trial showed TBWL of 7.1% (95% CI ± 5.0%) in the treatment group versus 3.6% (95% CI ± 5.1%) with the sham. There was one SAE secondary to a bleeding gastric ulcer [20,88]. The meta-analysis data suggest that air-filled balloons are better tolerated but have inferior WL [73,74]. However, two head-to-head RCTs that compared fluid-filled (Orbera) and air-filled (Heliosphere) balloons showed no significant difference in WL at six months, but a high self-deflation rate in air-filled balloons led to premature study termination [89,90]. 

### 4.2. Effect on Obesity-Related Complications

Several studies have assessed the impact of IGBs on obesity-related complications, including hypertension, T2D, and non-alcoholic fatty liver disease (NAFLD). In a meta-analysis of 5668 patients including 10 RCTs, an IGB for a maximum of six months was associated with a modest, but statistically significant, improvement in metabolic parameters, including fasting blood glucose, systolic blood pressure, HbA1c, alanine transaminase (ALT), and aspartate transaminase (AST) [91]. For NAFLD specifically, Lee et al. [92] showed no significant difference in histological NAFLD scores between Orbera balloon and sham procedure. However, Bazerbach et al. [93] showed a significant reduction in NAFLD activity scores based on liver biopsy in 21 patients treated with Orbera balloon in an open-label study. A meta-analysis of the Orbera balloon including 452 patients across nine studies showed a pooled improvement in NAFLD activity scores of 83.5% (95% CI 60.8–94.3) and steatosis on imaging in 79.2% (95% CI 66.3–88.1) [94]. The role of IGBs for the treatment of NAFLD is an emerging space but still limited by patient numbers. 

### 4.3. Combination Therapy

There has been interest in IGBs as a bridging therapy prior to bariatric surgery because preoperative WL is associated with decreased operative time and perioperative complications [95,96]. A multicentre RCT among 115 patients with a BMI > 45 kg/m^2^ undergoing RYGB compared standard preoperative care to an IGB (Orbera/Heliosphere) [97]. The balloon group lost significantly more preoperative weight at six months (−2.8 kg/m^2^ vs. −0.4 kg/m^2^; *p* < 0.0001) but had no difference in postoperative intensive care or hospital admission and higher 30-day complications. In a meta-analysis among 399 patients assessing preoperative IGBs for those with a BMI ≥ 50 kg/m^2^, balloon complications were reported in 8.13% (95% CI 4.04–13.17%) including two haemorrhages and one death from aspiration. Furthermore, among the prospective studies assessing surgical outcome the perioperative complication risk was higher [98]. 

There are limited data using IGBs in combination with novel pharmacological agents. One retrospective study suggested a higher TBWL% at 12 months when an IGB was combined with an anti-obesity drug; however, there was significant heterogeneity in drug choice [99]. A small retrospective study showed no benefit in preventing weight regain with liraglutide at six months following balloon removal [100]. High-quality prospective data on combination therapy are needed, specifically to address weight regain that has limited long-term efficacy.

## 5. Gastric Remodelling 

Over the last decade, endoscopic remodelling of the stomach has become an attractive primary WL intervention. Remodelling offers an effective, quick, and minimally invasive alternative to bariatric surgery associated with a lower number of AEs. There are two main devices currently available: Apollo OverStitch (Apollo Endosurgery, Austin, TX, USA) used for ESG and Primary Obesity Surgery Endoluminal (POSE) (USGI Medical, San Clemente, CA, USA). A third device called Endomina (Endo Tools Therapeutics, Charleroi, Belgium) has been CE marked and currently is being investigated in clinical trials but is not discussed further. 

### 5.1. Endoscopy Devices

#### 5.1.1. Apollo Overstitch

The Apollo system is a suturing device that attaches to a double-channel endoscope and applies full-thickness sutures to create a restrictive sleeve (Figure 2) [101]. It contains a needle driver, suture anchor, tissue helix, and actuating handle to transfer the suture. In 2012, the first-in-human ESG was performed in India [102], with the suture pattern modified at the Mayo clinic [103]. Different patterns can be used (‘U’ and ‘Z’) at the discretion of the endoscopist, as none have shown superior WL [104]. In 2017, Apollo gained FDA approval for their new Overstitch Sx system that can attach to a single channel endoscope with a broader range of endoscope compatibility [105]. This device is currently being evaluated through registry data and the SLEEVE trial (ClinicalTrials.gov: NCT05072067).

#### 5.1.2. Primary Obesity Surgery Endoluminal

POSE is a CE-marked endoscopic procedure that enables tissue apposition through creation of serosa-to-serosa plications with the g-Cath EZ Delivery Catheter and Incisionless Operating Platform. The original procedure placed 7–9 suture anchors in the gastric fundus to reduce volume and postprandial accommodation. An additional 3–4 sutures were placed in the distal gastric body to prolong satiety [106]. This was modified to target the gastric body with a perceived greater impact on motility, which became known as POSE-2 or ‘distal’ POSE. This involves placing 16–18 plications in symmetrical rows within the upper and lower part of the gastric body sparing the fundus [107].

### 5.2. Effect on Weight Loss

ESG has shown excellent short-term TBWL% in several meta-analyses that have reported 15.3–17.1% across 12–24 months follow-up and SAEs of 1.5–2.3% [108,109,110]. The recently published MERIT RCT involving 209 patients confirmed the safety and efficacy of ESG compared to lifestyle modification alone [9]. At 52 weeks, TBWL was 13.6% (SD 8.0) for ESG and 0.8% (SD 5.0) for control (*p* < 0.0001). After one year, the control group participants were offered ESG with no significant difference in the proportion achieving ≥25% EWL after 52 weeks (*p* = 0.21). SAEs occurred in three participants (2%), who were all managed non-surgically. Longer-term outcomes have been encouraging with Sharaiha et al. [111] reporting a five-year TBWL% of 15.9 (95% CI 11.7–20.5) among 38/216 patients, and a single centre in India reported a mean TBWL among 229 patients of 18.2% (95% CI 17.7–18.6) at four years [112]. Compared to IGB insertion, ESG has demonstrated superior WL with a mean difference of 3.1–11.5% over 6–24 months, and a lower SAE profile (1.5% vs. 4.0%) [113]. Compared to LSG, safety has remained similar, but efficacy has varied, with one study reporting superiority of LSG among 2188 over 12 months [114], whereas another reported non-inferiority of ESG among 3018 across 6–36 months [115].

Evidence for POSE is less conclusive as shown in two multicentre RCTs. In MILEPOST, 44 patients were randomly assigned to POSE or diet and exercise. The TBWL% in the POSE and control group was 13.0 (95% CI 10.3–15.8) and 5.5 (95% CI 0.3–10.3) at 12 months, respectively, with no SAEs [10]. In ESSENTIAL, 221 patients were randomized to POSE and 111 to sham. At 12 months, the TBWL% values in the active and sham groups were 4.95 (95% CI ± 7.4) and 1.38 (95% CI ± 5.58), respectively (*p* < 0.001), with SAEs in 5% [11]. These disappointing results led to development of POSE2.0 with the initial pilot conducted on 13 patients in 2017. The TBWL% at 12 months was 14.7 (95% CI ± 4.0) with no AEs [107]. The largest series to date has been among 75 patients with reported TBWL% values at six and 12 months of 17.5 (SD 6.5), and 17.8 (SD 9.5), respectively [116]. The rate of SAEs was 5% with no associated mortality. Follow-up studies have confirmed that 85% of patients (*n* = 44) achieved >10% TBWL at 12 months with no SAEs and improvements in hepatic steatosis [12].

### 5.3. Effect on Obesity-Related Complications

ESG has a positive impact on obesity-related complications including a significant reduction in hypertension, T2D, and hyperlipidaemia [9,117,118]. This effect appears to be non-inferior compared to conventional LSG [115]. Small prospective studies and registry data have demonstrated improvements in liver biochemistry and non-invasive fibrosis risk scores 1–2 years following ESG [119,120]. There are currently two RCTs assessing the benefit of ESG for non-alcoholic steatohepatitis (NASH) compared to LSG (TESLA-NASH, NCT04060368) or sham (NASH-APOLLO, NCT03426111) [121]. 

### 5.4. Gastric Remodelling Mechanisms

A delay in gastric emptying with an increase in satiety has been proposed to be the major cause of WL [122]. In a cohort of 17 patients from the MERIT trial, gastric emptying breath test showed that gastric emptying was delayed at three months compared to lifestyle intervention alone (152.3 min ± 47.3 vs. 89.1 min ± 27.9, *p* < 0.001), which remained at 12 months [123]. In the MERIT trial, 11 patients underwent MRI evaluation of the stomach that showed no significant effect on gastric motility at three and 12 months. Thus, by targeting the gastric body and preserving the fundus, it appears to create a reservoir for food that increases satiety due to delayed emptying. However, the results using POSE2.0 are more heterogenous. Among 36 patients undergoing gastric emptying studies using nuclear scintigraphy, after the procedure, 64% (23/36) of the patients had decreased emptying and 36% (13/36) of the patients had increased emptying [12]. A subcohort of 21 patients underwent breath testing that showed a delay in gastric emptying at two months that returned to baseline at six months.

We know from evaluation post LSG that wider changes occur in gut hormone regulation, bile acid signalling, and the microbiome [124]. LSG significantly reduces fasting ghrelin levels, which drives appetite through signalling in the hypothalamus [125,126,127]. It also leads to an increase in satiety hormones GLP-1 and PYY that may be induced by increased gastric emptying into the small bowel [128,129,130]. Few studies have evaluated the impact of gut hormones after gastric remodelling. Lopez-Nava et al. [131] found no difference in fasting ghrelin, GLP-1, or PYY after ESG in 12 patients, whereas Abu Dayyeh et al. [122] showed a decrease in fasting and postprandial ghrelin in 25 patients three months after ESG. Therefore, ESG may blunt the physiological rise in ghrelin seen after dieting. Conversely, Vargas et al. [123] demonstrated a significant increase in fasting ghrelin, GLP-1, and PYY at 18 months after ESG compared to baseline but without postprandial assessment. 

### 5.5. Combination Therapy

Very limited data are available on the combination of gastric remodelling and anti-obesity drugs. Currently, trials are ongoing to assess the efficacy of GLP-1 agonists following bariatric surgery (NCT05073835). Badurdeen et al. [132] looked at the combination of ESG plus liraglutide initiated five months post the procedure compared to ESG alone in 56 patients. TBWL% in ESG plus liraglutide at seven months was 24.7 (SD of 2.1) compared to 20.5 (SD of 1.7) with ESG alone (*p* < 0.001). There was no discontinuation and only mild AEs with liraglutide. 

Another question is the role of ESG before or after bariatric surgery. In patients who are high surgical risk, ESG has been proposed as a bridge to surgical intervention that was shown to be technically feasible and safe among 20 patients undergoing revisional LSG [133]. Revisional ESG, termed ‘sleeve-in-sleeve’, has been demonstrated to be safe and effective among 82 patients after LSG with TBWL of 15.7% at 12 months and a single stricture requiring balloon dilatation [33]. Meta-analyses have shown that both revisional ESG and LSG procedures can be performed safely after ESG [115]. This is critical given patients are living longer with obesity and will require additional options. 

## 6. Duodenal Mucosal Resurfacing

DMR is a novel technique that selectively ablates the duodenal mucosa to improve glycaemic control (Figure 3). It was developed after observations in the 1980s that diabetes dramatically improved following bariatric surgery [134,135]; the improvements were immediate and weight independent, and showed the importance of the proximal small bowel in insulin sensitivity. Subsequent RCTs have confirmed that surgical interventions that bypass the proximal small bowel are associated with a high, albeit heterogenous, rate of T2D remission that is reversible on re-exposure of nutrients via the remnant stomach [136,137,138,139]. 

Early preclinical studies on diabetic rat models have shown that disruption of the duodenal mucosa using abrasion devices lowers blood sugar [140]. The first in-human study on 39 patients with T2D showed a clinically significant reduction in HbA1c following a single procedure with the Revita system (Fractyl Laboratories) [141]. Haidry et al. [140] demonstrated that DMR was safe and effective in 29 patients who underwent ≥9 cm ablations. HbA1c was reduced after 24 weeks (9.7% ± 1.4 vs. 8.4% ± 1.9, *p* = 0.0008) with three cases of duodenal stenosis managed with balloon dilatation. The Revita system is the most well studied, but other technologies including steam and electroporation are in various stages of development in a rapidly expanding field. 

### 6.1. REVITA System

The Revita system consists of a catheter and console. The catheter is a sterile, single-use device that injects saline into the duodenal submucosa to create a thermal barrier prior to mucosal hydrothermal ablation using a balloon. The console is a reusable electro-mechanical device that helps in the functionality of the lift and ablation. The procedure is completed in ~70 min under general anaesthetic or conscious sedation.

### 6.2. Effect on Diabetes Mellitus

REVITA-1 was a prospective, open-label, multicentre feasibility study that assessed the REVITA system in non-insulin dependent T2D [24,142]. In total, 46 patients with HbA1c 59–86 mmol/mol and an oral hypoglycaemic agent underwent DMR. It could not be completed in nine patients due to technical challenges. At 24 weeks, HbA1c was reduced by 10 ± 2 mmol/mol (*p* < 0.001), and fasting plasma glucose was reduced by 1.7 ± 0.5 mmol/L (*p* < 0.001), which is akin to an additional hypoglycaemic agent. This effect was maintained at 12 months independent of WL. SAEs were observed in six (16.2%) patients with one procedure-related event (transient febrile illness). A 24-month uncontrolled extension among 27 patients showed sustained improvement in Hb1Ac, fasting glucose, and ALT with no device or procedure-related SAEs [142]. On follow-up, antidiabetic agents were increased in 24% of the patients, insulin was started in 12% of the patients, and remained unchanged in 50% of the patients, which is significant given 40% of the patients had an indication to start insulin at entry. 

REVITA-2 was a randomised, double-blind, sham-controlled, multicentre trial that enrolled 108 patients from sites in Europe and Brazil [25]. At 24 weeks, the combined median reduction in HbA1c was not significantly different (*p* = 0.15). However, when stratified for high fasting plasma glucose (>10 mmol/L), HbA1c reduction with DMR was significant (–14.2 mmol/mol vs. −4.4 mmol/mol, *p* = 0.002). A significant difference in liver fat percentage was only observed when stratifying for a baseline fat percentage >5% and fasting plasma glucose >10 mmol/L. In a post hoc analysis that separated European and Brazilian cohorts, a significant difference was noted in the European cohort (−6.6 mmol/mol vs. −3.3 mmol/mol, *p* = 0.033). There were two SAEs, i.e., a haemorrhoidal bleed and a jejunal perforation requiring surgery. The duodenum appeared normal with complete healing on follow-up endoscopies. The REVITA-2 trial showed that DMR can have a positive treatment effect on metabolic health in the correct populations. 

### 6.3. Experience beyond Diabetes 

There is limited experience outside of the REVITA trials on DMR. The DOMINO trial was a double-blinded RCT on DMR versus sham among women (*n* = 30) with polycystic ovary syndrome and insulin resistance [143]. There was no significant difference in insulin sensitivity at 12 and 24 weeks; however, the patients had minimal metabolic dysfunction at baseline with a mean HbA1c of 39.7 mmol/mol (±3.7). 

### 6.4. Combination Therapy

In a small (*n* = 16) prospective pilot study, the combination of DMR (single session) with the GLP-1 agonist liraglutide allowed 69% of the patients to come off insulin with a HbA1c ≤ 7.5% at six months, which was maintained in 53% of the patients at 18 months [144]. An interesting concept for the future will be combining gastric remodelling with DMR. 

## 7. Transoral Outlet Reduction

Weight regain and dumping syndrome (DS) are common complications associated with RYGB, which are linked to the gastrojejunal anastomosis (GJA). GJA dilatation is an independent risk factor for weight regain and can drive rapid emptying of the pouch promoting DS [145,146]. Surgical revision is technically challenging, invasive, and associated with significant morbidity [147]. This has led to TORe, which is a type of endoscopic day-case revisional procedure, that utilises the Apollo Overstitch to tighten the dilated GJA with non-interrupted sutures in a purse string pattern [148].

### 7.1. Effect on Weight Loss

There is now >10 years of evidence to support TORe. In 2013, the RESTORe RCT, which involved 77 patients, showed a significant TBWL% at 6 months compared to a control (3.5% vs. 0.4%, *p* = 0.021) [30]. A retrospective review of 123 patients undergoing TORe who reached five years of follow-up showed a TBWL of 8.8% (95% CI ± 12.5%) and no SAEs [149]. Another group showed a TBWL of 8.1% (95% CI 3.1–13.3) among 56 patients reaching 24 months follow-up [150]. Meta-analysis data of 330 patients showed a TBWL of 8.4 kg (95% CI 6.5–10.3) at 12 months [31]. Finally, compared to revisional surgery, there was similar TBWL% at 1, 3, and 5 years with a lower AE rate (6.5% vs. 29.0%, *p* = 0.04) [151]. 

### 7.2. Effect on Dumping Syndrome

DS is a postprandial phenomenon characterised by vasomotor symptoms. The efficacy of any intervention is measured by the Sigstad score that can determine severity. In a study of 115 patients, TORe was associated with a significantly improved Sigtad score (17 ± 6.1 to 2.6 ± 1.9, *p* = 0.0001) [152]. This is consistent with meta-analysis data among 263 patients who had a mean Sigstad score difference of minus 9.96 points. There were no SAEs and reintervention was successful in >70% of the ~1/10 who had recurrent symptoms [153].

## 8. Future Perspectives

Obesity is fast becoming the most significant health concern of the 21st century. There is an urgent need for safe, effective, and scalable interventions that can be offered to patients of all different weight categories. The introduction of ‘metabolic endoscopy’ offers an ever-increasing armamentarium of obesity treatment procedures and devices. However, its introduction is not simple as the procedures often require a high degree of technical ability, and it is not clear how they will integrate into advanced endoscopy programmes. While evidence is mounting for the safety and efficacy of the devices, most are not NICE approved in the UK, which makes access challenging. Before discussing the future of metabolic endoscopy, we must address the current obesity crisis and why metabolic endoscopic interventions are urgently needed. 

Non-invasive lifestyle interventions are the first-line option for overweight and obesity, but such interventions are failing to address the current crisis. It is estimated that the annual rate of weight normalisation among obese individuals is less than 1% [154]. This means most individuals with obesity remain obese during their adult lifetime. There are multiple factors that contribute to these figures including access to healthy food, socioeconomic status, portion sizes, food education, and misinformation among others that form the modern obesogenic environment [155]. Dieting programmes are promoted widely and can achieve short-term WL, but the vast majority lead to weight regain within the first year [156,157,158]. In fact, restrained eating and exercise for weight control are predictors of weight gain [159]. Conventional hypocaloric diets have become the prototypical approaches that reduce daily energy intake by 500–750 kcal; however, individual responses are variable and they are hard to maintain long term [160]. Part of the problem is due to metabolic compensation, in which energy expenditure falls in response to energy deficit, which is a biological adaptation to prevent further WL associated with increased levels of ghrelin and reduced anorexigenic hormones [157]. Metabolic adaptations may persist for a year after WL, which is why weight regain is so common [40]. Consequently, the complex hormonal regulation between brain and gut that governs metabolic health has become a target for drugs and endoscopic intervention. 

Novel anti-obesity drugs are proving to be highly effective as WL treatments. They replicate gut hormone responses that promote satiety, reduce food intake, and prevent weight gain. However, whether patients can sustain WL without medications long term is yet to be determined, with early results showing weight regain on withdrawal [65]. Drugs are associated with side effects, require concordance, involve injections, and have unknown long-term consequences. We may see them mirror those in T2D, where sequential medications are needed leading to a lifetime of polypharmacy. Consequently, endoscopic interventions offer an exciting alternative to life-long drug maintenance. 

IGBs were the earliest devices that have shown consistent short-term TBWL between 12 and 15% at six months. However, long-term efficacy of IGBs is lacking and given the not insignificant AEs rate, their efficacy seems akin to high intensity dietary changes in the short term. Their use as bridging therapies has thus far given poor results. How they fit into the modern hierarchy of treatments is unclear, but we cannot underestimate the value of ‘procedureless’ balloons from a patient perspective. IGBs could offer an alternative at lower BMI in combination with high-intensity diet and exercise, or a surrogate marker of tolerance/response to gastric remodelling. 

ESG using the Apollo Overstitch has consistently shown 13–17% TBWL at one year in RCT and meta-analysis data that appear durable over five years. It has an excellent safety profile (SAEs of 1–3%) which is consistent with most therapeutic endoscopic procedures. Mechanistic data are limited, but ESG does appear to slow gastric emptying and prevent the compensatory rise in ghrelin that makes it hard to maintain traditional hypocaloric diets. Blunting this compensatory rise makes it an excellent strategy to combine with GLP-1 agonists. The Overstitch requires a degree of technical ability that limits its widespread use. However, endoscopic suturing has broad applications within the GI tract, including revisional endoscopy after bariatric surgery (e.g., TORe), which means it will be essential for all advanced endoscopists. 

DMR is in its infancy, but like bariatric surgery, it targets the root problem of diabetes. Insulin resistance is driven by a complex and incompletely understood interaction between small bowel nutrients and whole-body hormone signalling. By altering small bowel mucosa, early data have shown HbA1c reduction by 6–14 mmol/mol in a weight-independent manner. This helps to prevent initiation of insulin that often leads to a vicious cycle of weight gain, worsening glycaemia, and increasing insulin doses. DMR offers a one-off intervention to break this cycle and thus improve metabolic health, which could be repeated. Currently, the device has a high technical barrier, but as more accessible devices become available it will undoubtedly become more prominent.

One of the main challenges in metabolic endoscopy is to whom it should be offered. Bariatric surgery is the most effective treatment for patients with severe obesity (BMI ≥ 40 kg/m^2^). However, at lower BMI values the current dogma is diet and exercise until patients develop worsening BMI or obesity-related complications. This means that many patients remain obese for years without an effective treatment. Given that so few patients ever regain normal weight during their lifetime, there is an argument for a ‘top-down approach’ at early obesity levels. This means offering endoscopy with or without drug therapy at a much earlier stage, which is akin to modern treatment of autoinflammatory conditions where the disease is controlled with high-intensity pharmacotherapy before subsequent de-escalation. This will require defining indications and treatment targets at a national level.

Any future application of metabolic endoscopy must consider its integration into endoscopy and gastroenterology training. Obesity is a major risk factor for many GI diseases, which means modern metabolic gastroenterologists could treat a wide range of conditions including Barrett’s, gastro-oesophageal reflux, gallstones, NAFLD, irritable bowel syndrome, and even T2D. In the UK, advanced fellowships in therapeutic endoscopy must incorporate metabolic techniques, including management of bariatric complications. Ultimately, metabolic endoscopy is a rapidly growing field that offers patients effective, one-off interventions, as an alternative to bariatric surgery. Given the dramatic rise in obesity, we feel there is a priority to grow and develop this novel subspeciality, which will undoubtedly see an explosion as devices become NICE approved. 

## Figures and Tables

**Figure 1 life-13-01905-f001:**
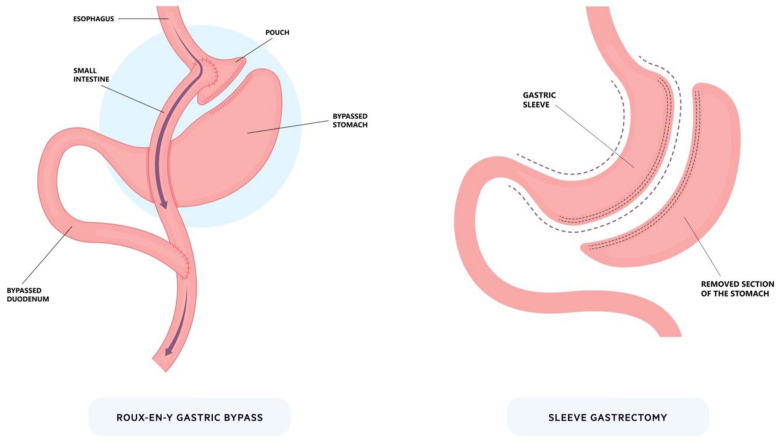
Bariatric surgery. The two most prominent bariatric procedures: Roux-en-Y gastric bypass and sleeve gastrectomy. For Roux-en-Y, a small pouch is created from the stomach that is connected to the intestines to bypass part of the proximal upper gastrointestinal tract. In sleeve gastrectomy, a large proportion of the stomach is removed (~80%) along the greater curvature to significantly reduce its size.

**Figure 2 life-13-01905-f002:**
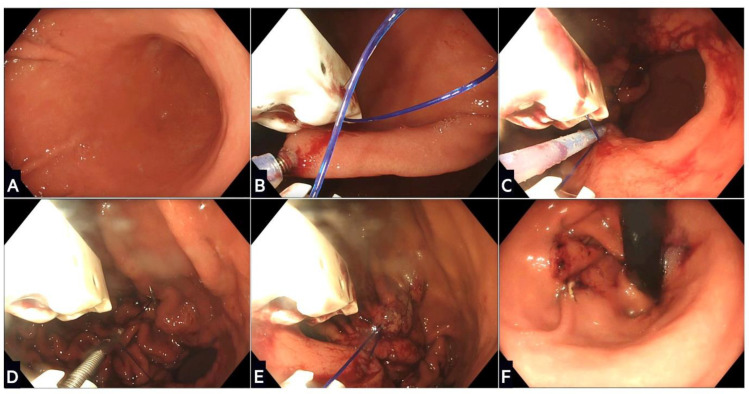
Stages of endoscopic sleeve gastroplasty using the Apollo OverStitch: (**A**) The stomach is washed and a location proximal to the antrum identified; (**B**) gastric tissue is grasped and pulled towards the device using the helix, and then a full thickness suture is placed; (**C**) further sutures are placed in a ‘U’ pattern starting from the anterior wall, moving along the greater curvature, and towards the posterior wall; (**D**) the suture is this then tightened and locked in a process called cinching; (**E**) the process is repeated with two or three further rows of sutures placed proximally to create the restrictive sleeve; (**F**) the final appearance of the remodelled stomach in a retroflexed scope position.

**Figure 3 life-13-01905-f003:**
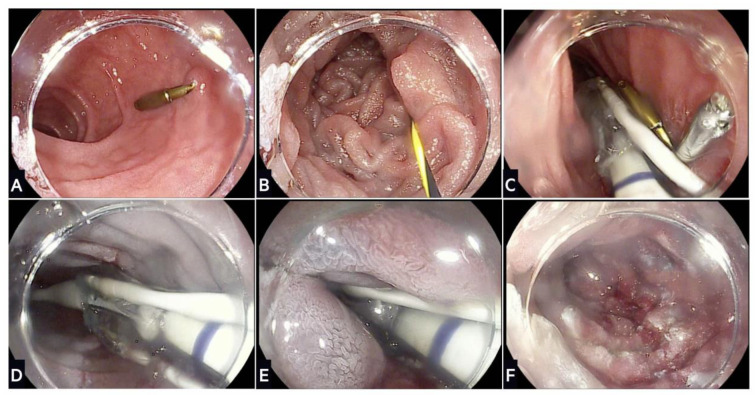
Stages of duodenal mucosal resurfacing using the Revita device: (**A**) After screening gastroscopy to ensure no contraindication, the ampulla of Vater is located and an endoscopic clip placed on the contralateral wall; (**B**) a guidewire is inserted through the duodenum beyond the ligament of Treitz, which is used to advance the DMR catheter; (**C**) fluoroscopy is used to confirm positioning of the guidewire and catheter during the procedure; (**D**) a submucosal lift is performed that creates a protective saline layer with the submucosa; (**E**) stepwise circumferential hydrothermal ablation is conducted at 90 °C for ~10 s starting just distal to the ampulla of Vater and continuing for 9–10 cm distally; (**F**) final appearance of the duodenal mucosa following hydrothermal ablation.

**Table 1 life-13-01905-t001:** A summary of the safety and efficacy of the most prominent metabolic and bariatric endoscopic procedures.

Gastric								
Endoscopic Option	Procedure	Description	Key Trials	Trial Type	No. Patients	WL 4–6 Months (SD)	WL 12 Months (SD)	SAE
Gastric remodelling	ESG (Apollo Endosurgery)	Suturing to reduce stomach volume	MERIT [9]	RCT	209	-	TBWL 13.5% (±8.0)	2.0%
	POSE 1.0 (USGI medical)	Endoscopic plications to reduce stomach volume (fundus)	MILEPOST [10] ESSENTIAL [11]	RCT RCT	44 332	- -	TBWL 13.0% (±2.7) TBWL 4.95% (±7.0)	0.0% 5.0%
	POSE 2.0 (USGI Medical)	Endoscopic plications to reduce stomach volume (body)	Lopez Nava et al. [12]	PRO	44	-	TBWL 15.7% (±6.8)	0.0%
	Endomina (Endo Tools Therapeutics)	Endoscopic suturing to reduce stomach volume	Huberty et al. [13]	RCT	71	-	TBWL 11.9% (±2.6)	0.0%
Gastric balloons	Orbera (Apollo Endosurgery)	Fluid-filled balloon inserted by endoscopy	Kumar et al. [14] ASGE [15]	MA MA	5549 1638	TBWL 13.2% (±0.8) -	- EWL 25.4% (±4.0)	1.5%
	Spatz (Spatz Medical)	Fluid-filled balloon inserted by endoscopy	SABO [16]	RCT	288	TBWL 15.0% (±1.1)	-	4.0%
	Elipse (Allurion Technologies)	Fluid-filled balloon inserted by swallowing	Ramai et al. [17] Jamal et al. [18]	MA PRO	2152 112	TBWL 12.0% (±1.2) -	- TBWL 7.9% (±6.7)	0.5% 0.0%
	Heliosphere (Helioscopie)	Air-filled balloon inserted by endoscopy	Lecumberri et al. [19]	PRO	84	TBWL 13.4% (±7.0)	-	0.0%
	Obalon (Obalon Therapeutics)	Air-filled balloon inserted by swallowing	SMART [20]	RCT	387	TBWL 7.1% (±5.0)	-	0.4%
	Reshape Duo (Apollo Endosurgery)	Fluid-filled balloon inserted by endoscopy	REDUCE [21]	RCT	326	EWL 25.1% (±1.6)	-	3.0%
Transpyloric shuttle	Transpyloric shuttle (BAROnova)	Spherical bulb attached to a smaller bulb across the pylori	ENDObesity II [22]	RCT	302	-	TBWL 9.5% (±0.7)	4.7%
Gastric aspiration	AspireAssist (Aspire Bariatrics)	Gastrostomy tube used to aspirate gastric content	PATHWAY [23]	RCT	207	-	TBWL 12.1 (±9.6)	3.6%
**Small intestinal**								
**Endoscopic option**	**Procedure**	**Description**	**Key trials**	**Trial type**	**No. patients**	**Mean fall in HbA1c (24–34 weeks)**		**SAE**
Duodenal mucosal resurfacing	Revita (Fractyl)	Hydrothermal ablation of the duodenal mucosa	REVITA-1 [24] REVITA-2 [25]	PRO RCT	46 108	−10.0 mmol/mol −10.4 mmol/mol		2.2% 3.6%
	Electroporation Therapy (Endogenex)	Electroporation of the duodenal mucosa	EMINENT * [26]	PRO	14	−6.6 mmol/mol		0.0%
Bypass liners	Endobarrier (GI Dynamics)	60 cm Impermeable liner anchored in the duodenum	Koehestanie et al. [27] Jirapinyo et al. [28]	RCT MA	73 412	−13.3 mmol/mol −13.3 mmol/mol		14.7% 15.7%
Partial jejunal diversion	Incisionless anastomotic system (GI Windows)	Self-assembling magnets placed between jejunum and ileum	Machytka et al. [29]	PRO	10	−18.6 mmol/mol		10.0%
**Revisional**								
**Endoscopic option**	**Procedure**	**Description**	**Key trials**	**Trial type**	**No. patients**	**WL 6 months (SD)**	**WL 12 months (SD)**	**SAE**
Revision gastrojejunal anastomosis after RYGB	TORe (Apollo Endosurgery)	Endoscopic suturing to narrow the gastrojejunal anastomosis	RESTORe [30] Vargas et al. [31]	RCT MA	77 330	TBWL 3.5% (±1.7) -	- WL 8.4 kg (±2.5)	0.0% 0.0%
	APC (Olympus)	APC to narrow the gastrojejunal anastomosis	Gurian et al. [32]	RCT	66	TBWL 7.5% (±3.6)	TBWL 1.8% (±5.7)	2.6%
Revision sleeve gastrectomy	ESG: sleeve-in-sleeve (Apollo Endosurgery)	Endoscopic suturing to reduce stomach volume	Maselli et al. [33]	PRO	82	-	TBWL 15.7% (±7.6)	1.2%

* Poster presentation. APC—Argon photocoagulation; ASGE—American Society of Gastrointestinal Endoscopy; ESG—endoscopic sleeve gastroplasty; MA—meta-analysis; No—number; POSE—primary obesity surgery endoluminal; PRO—prospective; RCT—randomised-controlled trial; RYGB—Roux-en-Y gastric bypass; SAEs—serious adverse events; SD—standard deviation; TBWL—total body weight loss; TORe—transoral outlet reduction endoscopy; WL—weight loss.

**Table 2 life-13-01905-t002:** A summary of the key characteristics of commercially available intragastric balloons.

Balloon Type	Filling	Insertion	Removal	FDA Approval	CE Mark	Time In Situ (Marks)
Orbera (Apollo Endosurgery)	Liquid	Endoscopy	Endoscopy	+	+	6 *
Spatz (Spatz Medical)	Liquid	Endoscopy	Endoscopy	+	+	12
Elipse (Allurion Technologies)	Liquid	Swallow	Excretion	-	+	4
Heliosphere (Helioscopie)	Gas	Endoscopy	Endoscopy	-	+	6
Obalon (Obalon Therapeutics)	Gas	Swallow	Endoscopy	+	+	6

* The Reshape^®^ Duo balloon was bought by Apollo and taken off the market. Orbera have launched a 12-month balloon.

## Data Availability

This was a literature review of previously published data. All articles are available through EMBASE and MEDLINE.

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
