# Peer review of "Metabolic and Bariatric Endoscopy: A Mini-Review"

_life, 2023, doi:10.3390/life13091905_

Round 1

Reviewer 1 Report

The paper is covering probably majority of techniques of endoscopy which try to eliminate bariatric surgery because of it cost and inconvenience for patients.

I would see the paper storage sorted out with more deep approach.

It looks like that now body would like to work with such approach. In fact bariatric surgery genuine discovery was that pancreatic endocrine and exocrine functions are governed mainly from duodenum.  

Modern, overflowing with “easy get and cheap” food eating behaviors affect both exo- as well as endo-pancreas. Thus, surgical elimination of pancreatic enzymes as well as surgical limitation of food is minimizing amount of elementary nutrients which need to be process after. In sportive nations absorbed elementary nutrients they are utilized to energy and body building, while in “coach potato” nations they are converted to fat. Both lates actions are endo-pancreatic hormones dependent. Paradox, dietary fat will the last one to be converted to host fat – glucose and amino acids are leading in that process.

Now coming the essential statement/question for bariatric endoscopy or surgery. How, much insulin can pancreas produce -think about it when injecting Ozempik. Especially if obese host develop insulin resistance. That number of insulin produced is limited, obese people develop DT2 and after DT1. All manipulation with endoscope or regular bariatric surgery acting against DT2 you described. They are protecting pancreas from exhausting and destroying    of beta -cells when protective mechanism – insulin resistance is to weak to stop eating and host is consuming overdoses of nutrients even with strong hyperglycemia.

I would like to see something like above, the green line as above why you write this review.

In the end paper is OK with actually prepared review standards.

Author Response

Dear reviewer,

Thank you for taking the time to go through our manuscript and make valuable comments. We appreciate your opinion that the paper should have a section outlining the crux of the problem in relation to the pathophysiology of obesity and metabolic syndrome, which has given rise to these minimally invasive endoscopic procedures.

Consequently, we have added a section termed ’The pathophysiology of obesity and weight regain’ that discusses the physiological mechanisms involved in weight gain and the central role of insulin. It goes on to discuss the process of insulin resistance, the difficulties with dietary manipulation, and the success of bariatric surgery that has paved the way for endoluminal approaches.

We believe this now sets the tone of the paper for an in-depth discussion on the treatment strategies for obesity focusing on metabolic and bariatric endoscopy.

BW,

Ben

Reviewer 2 Report

Dear Author(s),

Thank you for your interesting review article on metabolic and bariatric endoscopy.

The world is grappling with the increasing rates of obesity and T2D; thus, this is definitely a valuable and clinically relevant manuscript. 

To the best of my knowledge, the review article is well-summarized, easy to read and well-organized within the (sub)paragraphs. 

Regarding pharmacological paragraph, please also mention liraglutide as an already registered anti-obesity medicine. Also within line 96, you mention side-effect, which is possible and true, but they are not serious. Also endoscopy may have side-effects and complications so I would suggest you to modify the term or sth. within the sentence. 

When mentioning cost-effectiveness within the text please cite the relevant source (ref.) that demonstrates the cost-effectiveness. 

Consider adding data on swallowable capsules (with intragastric balloons) in more details; since this is a promising field that is rapidly developing (e.g. Allurion).

Check if you commented on all the endoscopy procedures available. Here is also a well-summarized paper (you used it as ref. 23).

Great Future perspectives paragraph for sure.

Nice work. Compliments.

I strongly believe that after this minor corrections your manuscript would be ready for publishment.

Author Response

Dear reviewer,

Thank you for taking the time to go through our manuscript and make valuable comments. We really appreciate your kind words and have taken on board the changes you have suggested. Below, please find a point-by-point response to your comments.

Comment 1:

Regarding pharmacological paragraph, please also mention liraglutide as an already registered anti-obesity medicine. Also within line 96, you mention side-effect, which is possible and true, but they are not serious. Also endoscopy may have side-effects and complications so I would suggest you to modify the term or sth. within the sentence.

Response 1:

We have added detail about the two main licensed anti-obesity medications (Liraglutide and semaglutide) into the paragraph 3.2. pharmacological. Within this with have also touched on adverse effects with a more neutral stance to avoid being biased towards endoscopic therapies.

Comment 2:

When mentioning cost-effectiveness within the text please cite the relevant source (ref.) that demonstrates the cost-effectiveness. 

Response 2:

We have added reference ’66’, which refers to a cost-effectiveness model completed in the UK comparing endoscopic sleeve gastroplasty to lifestyle modification alone. We have also included reference ’67’ that was a cost-effectiveness analysis completed in America comparing endoscopy, surgery, and pharmacotherapy for obesity. These can be found in paragraph 3.3 Endoscopy.

Comment 3:

Consider adding data on swallowable capsules (with intragastric balloons) in more details; since this is a promising field that is rapidly developing (e.g. Allurion).

Response 3:

We did briefly discussed the swallowable Allurion balloon in paragraph 4.1. weight loss. This discussed the efficacy of all major gastric balloons. We have now expanded on the detail for the Allurion balloon in this section based on current evidence.

Comment 4:

Check if you commented on all the endoscopy procedures available. Here is also a well-summarized paper (you used it as ref. 23).

Response 4:

With a large number of endoscopic devices past and present being investigated for their effect on weight and glycemic control, we chose to focus on only the most prominent procedures. We are clear about this in the background section of the manuscript (i.e. the mini-review is not meant to be an exhaustive review of every procedure). We have already summarised a large number of devices and many of those commented on in reference ’23’ are now obsolete. Therefore, we have opted to stick with only the procedures we list in the manuscript.

Once again, thank you for taking the time to review our manuscript and providing insightful comments and suggestions.

BW,

Ben